# Housing Insecurity and Other Syndemic Factors Experienced by Black and Latina Cisgender Women in Austin, Texas: A Qualitative Study

**DOI:** 10.3390/ijerph20247177

**Published:** 2023-12-13

**Authors:** Liesl A. Nydegger, Erin N. Benton, Bree Hemingway, Sarah Fung, Mandy Yuan, Chau Phung, Kasey R. Claborn

**Affiliations:** 1Department of Health, Behavior & Society, Johns Hopkins Bloomberg School of Public Health, Baltimore, MD 21205, USA; 2Department of Kinesiology & Health Education, The University of Texas at Austin, Austin, TX 78712, USA; enbenton@me.com; 3School of Community & Global Health, Claremont Graduate University, Claremont, CA 91711, USA; bree.l.hemingway@cgu.edu; 4Moody School of Communication, The University of Texas at Austin, Austin, TX 78712, USA; sarahfung00@gmail.com; 5School of Human Ecology, The University of Texas at Austin, Austin, TX 78712, USA; m_yuan@utexas.edu; 6Department of Neuroscience, The University of Texas at Austin, Austin, TX 78712, USA; chaup@utexas.edu; 7Steve Hicks School of Social Work, The University of Texas at Austin, Austin, TX 78712, USA; kasey.claborn@austin.utexas.edu

**Keywords:** Black/African American cisgender women, Latina cisgender women, housing insecurity, intimate partner violence, substance use, HIV risk, qualitative research

## Abstract

Austin, Texas emerged as one of the fastest-growing cities in the U.S. over the past decade. Urban transformation has exacerbated inequities and reduced ethnic/racial diversity among communities. This qualitative study focused on housing insecurity and other syndemic factors among Black and Latina cisgender women (BLCW). Data collection from 18 BLCW using in-depth interviews guided by syndemic theory was conducted three times over three months between 2018 and 2019. Four housing insecurity categories emerged: (a) very unstable, (b) unstable, (c) stable substandard, and (d) stable costly. Participants who experienced more stable housing, particularly more stable housing across interviews, reported fewer instances of intimate partner violence (IPV), less substance use, and a reduced risk of acquiring HIV. Results identified the importance of exploring housing insecurity with other syndemic factors among BLCW along with determining structural- and multi-level interventions to improve housing circumstances and other syndemic factors. Future research should explore these factors in other geographic locations, among other intersectional communities, and among larger sample sizes and consider using a mixed methods approach.

## 1. Introduction

Austin, Texas has grown by over 29% between 2010 and 2019 [1], making it one of the fastest growing cities in the U.S. [2], which has led to exponential growth and gentrification, exacerbating inequities, and reducing ethnic and racial diversity in communities [3]. This provides a unique opportunity to better understand housing insecurity and other syndemic factors, such as intimate partner violence (IPV), substance use, and the risk of acquiring HIV within the context of housing insecurity in a large urban area. Housing insecurity encompasses a number of challenges, such as overcrowding, difficulty paying rent, spending the majority of the household income on housing, and moving frequently. All of these challenges can greatly affect one’s physical and mental health [4,5].

From 2018 to 2019, when data were collected, the population of Austin had an annual growth rate of 3.66% [6]. Black/African American populations are the only ethnic minority group that decreased in Austin due to these populations moving away from the city center and having a higher mortality rate [7]. In 2022, 7.7% of Black/African American and 33.1% of Latino/a communities comprised the population of Austin, TX [8]. Of those who experienced homelessness in Austin, over one-third (37%) were Black/African Americans and 32% were Latino/as [9]. Of individuals experiencing homelessness, 40% were women, 28.3% experienced IPV, and 19.9% were children [10]. Nationally, among individuals experiencing homelessness, 39.4% were Black/African American, 22.5% were Latino/a, 38.5% were women, and 18.3% were children [11]. Although there are low populations of Black/African Americans in Austin, these communities experience high rates of homelessness, and Latino/a communities and women and children experience higher rates of homelessness in Austin compared to the U.S. Housing assistance is available in Austin, but typically has a long waiting list or is difficult to access. Options include emergency voucher programs, a Housing Choice Voucher program (formally known as Section 8), and housing provided by local non-profits [12,13,14]. Additionally, Black and Latina cisgender women (BLCW) in Austin reported coinciding syndemic factors [15]. Therefore, it is important to explore syndemic factors within the context of housing insecurity among BLCW with children in Austin.

### 1.1. Syndemic Model

Syndemic theory posits that conditions, such as structural, social, and individual-level factors, are mutually reinforcing and interact leading to adverse health outcomes (see Figure 1) [16,17]. Previous studies found that housing insecurity, such as a lack of access, affordability, and homelessness were related to IPV [18,19,20], substance use [16,21,22], and factors that increased one’s risk of acquiring HIV [23,24,25,26]. There is a dearth of research that has explored housing among BLCW through a syndemics lens. Considering the heightened risk of housing adversity among BLCW, research is urgently needed to better understand factors that are associated with housing to inform interventions for this high-risk, underserved population.

### 1.2. Study Purpose

Limited research has focused on housing insecurity and other syndemic factors among BLCW at a high-risk of acquiring HIV. As of 2018 when the study began, 29% of the total U.S. population experiencing homelessness were women [27]. Potential health consequences of homelessness include increased rates of diabetes, hypertension, HIV, substance use disorders, and mental illness. People who experience homelessness have higher rates of illness and die on average 12 years sooner than the general U.S. population [28]. The present qualitative study sought to explore housing experiences in-depth and across time among BLCW. Additionally, we considered other factors that reinforce and/or interact with housing, including IPV, substance use, and risk factors for acquiring HIV. The guiding research question was, “How was housing insecurity impacted by and impacted other syndemic factors, particularly intra- and inter-personal factors, among BLCW in Austin, TX?”.

## 2. Materials and Methods

### 2.1. Participants

This study is an analysis of a larger qualitative study exploring barriers to pre-exposure prophylaxis (PrEP) among BLCW who were at a high risk of acquiring HIV [15]. Syndemic theory guided the development of study materials. During an analysis of the larger study, we determined that housing insecurity was frequently reported by participants and thus is the focus of this manuscript. We conducted longitudinal, in-depth, semi-structured interviews at three separate assessment points (baseline, 1-, and 3-months) among 18 BLCW (*M age* = 31.8; *SD* = 7.55; range 21–47 years) at-risk of acquiring HIV from May 2018 to November 2019. The main, larger study was a formative, observational study to ascertain syndemic factors that impacted the risk of acquiring HIV and PrEP interest and adoption to develop future interventions. The baseline interview guide explored syndemic factors both prior to and currently experienced by participants, interview guide 2 explored changes in syndemic factors from the baseline interview, and interview guide 3 explored changes in syndemic factors from the 1-month interview and suggested future interventions. At the end of the first interview, participants were provided information about PrEP and PrEP providers. We interviewed participants again one month later to determine if they sought out a PrEP provider and three months later to determine if they adopted PrEP and if not, to identify syndemic barriers (see Ref. [15] for results regarding PrEP, barriers, and suggested interventions). Fourteen participants completed three interviews across three months; one participant completed two interviews and three participants completed one interview. Prior to signing the informed consent, we assessed preliminary eligibility for the larger study over the phone, which included (1) being at least 18 years, (2) identifying as BLCW, (3) having at least one child under 18 years, (4) having unprotected vaginal or anal sex with a cisgender man in the past 30 days, (5) having an HIV-negative or unknown status, and (6) spoke fluent English or Spanish. One participant requested that all three of her interviews were conducted in Spanish. After signing the informed consent, participants completed a structured interview regarding the remaining eligibility criteria, including participants who had at least two of the following experiences: (1) IPV in the past three months [29,30], (2) substance misuse in the past 30 days [31,32,33,34], (3) transactional sex in the past three months [31], (4) multiple sex partners in the past three months [31,35], and/or (5) their partner(s) had multiple sex partners in the past three months [15,31,35]. Exclusion criteria included (1) being less than 18-years-old, (2) identifying as any race or ethnicity aside from Black/African American or Latina/Hispanic, (3) identifying as any gender other than a cisgender woman, (4) having no children or children older than 18-years-old, (5) using condoms during every vaginal or anal sexual encounter with a cisgender man in the past 30 days, (6) living with HIV, (7) only spoke fluently in any language other than English or Spanish, and (8) having no access to a phone to determine preliminary eligibility.

### 2.2. Procedures

The Institutional Review Board at The University of Texas at Austin approved all study procedures. Recruitment occurred based on posting flyers at community organizations, local clinics, and around the City of Austin and incentivized snowball sampling. We used these sites to recruit BLCW from a variety of places to ensure a diverse background of participants. Participants were given the option to refer up to three eligible participants, receiving $10 for each eligible referral. Initial eligibility occurred over the phone and if preliminarily eligible, an in-person interview was scheduled. Interviews occurred in a private place that the participants preferred, including their homes, parks, or an interview room.

Once in person, participants were provided the informed consent and then a contact information sheet to keep in touch throughout the study. After the informed consent was signed, the audio-recorded structured interview began to ensure additional eligibility (see Ref. [15] for in-depth procedures). All interviews were recorded using two audio recorders with microphones. We used two in case one recorder malfunctioned to ensure that we obtained the recording. Participants who were ineligible were paid $5 for their time. Eligible participants continued with the in-depth, semi-structured interview and were interviewed again at one and three months after baseline.

Participants were contacted every two weeks in between interviews to increase study retention. Participants were compensated with increasing amounts of $25, $30, and $40 for interviews; those who traveled to the interviewer’s office were reimbursed $5 and those who needed childcare during interviews were reimbursed $15. All interviews were conducted in English (*n* = 17) or Spanish (*n* = 1) by female interviewers. The main interviewer was trained in qualitative methods and implemented previous qualitative studies [15,16,36,37,38] conducted in a majority of interviews. The remaining interviews were conducted by a research assistant trained by the main interviewer who had experience in social work and conducting qualitative interviews and a Spanish-speaking research assistant who had experience conducting qualitative research in Spanish in South America. Interviews were between 30 min and 3.3 h (*M* = 1.4 h). Interview lengths depended on the length of the interview guide (i.e., interview guides 1 and 3 were the longest) and participant responses. For example, if participants reported no substance use prior to that interview, substance use-related questions were skipped. Participants were informed about the sensitive nature of many of the questions so that they could choose the most appropriate interview location that was quiet and had no one around.

### 2.3. Data Collection

Interview guides explored daily life experiences (i.e., daily routine, economic situation), housing (i.e., current and previous living situations), relationships and romantic experiences, IPV, substance use, risk of acquiring HIV, and HIV-related knowledge and attitudes. When enquiring about IPV, interviewers stated the following: “No matter how well a couple gets along, there are times when they disagree, get annoyed with the other person, want different things from each other, or just have spats or fights because they are in a bad mood, are tired, or for some other reason. Couples also have many different ways of trying to settle their differences. I’m going to list things that might happen when you have differences. Please let me know if any of these happened to you in the last 3 months with your partner(s)”. Next, rather than asking participants if they experienced each type of IPV, interviewers described different forms of IPV that were adapted from the CTS-2 and Checklist of Controlling Behaviors (e.g., physical: “You had a sprain, bruise, small cut, or felt pain the next day because of a fight with your partner”; emotional: “Your partner destroyed something that belonged to you”; sexual: “Your partner insisted on sex without a condom when you wanted to use one”) [29,30]. Participants were also asked to tell the interviewer about the last fight or argument with their partner, and the interviewer probed for details, including how the fight started, who else was present, what occurred during the fight, and what occurred afterward. Regarding sexual coercion or violence, participants were also asked, “Have you ever had sex with anyone when you didn’t want to?” and “Have you ever had sex and felt bad about it afterwards?” If participants responded “yes,” to either or both questions, the interviewer stated, “If you are comfortable, please tell me about that situation”. For substance use, participants were initially asked how many days in the last 30 days that they drank alcohol and which type(s) of alcohol. They were then given a definition of 1 drink for each type of alcohol and asked how many drinks they would have in a typical week, and when they last consumed four or more drinks on one occasion. Next, interviewers listed different illicit drugs and marijuana, and if participants stated that they consumed any illicit drug or marijuana in the past 30 days, they were asked how many days. Later on, participants were asked about substance use again, clarifying what they stated before, and participants were also asked to describe instances when they used each substance. Interviewers would probe for details regarding how participants’ felt prior to using each substance and during, who they used substances with and on what occasions, and what other things tended to occur while using each substance (e.g., talking, arguing, dancing, sex). Participants were also asked about factors that increase their risk of acquiring HIV, such as how many partners and what types of partners (e.g., main, casual) they had, how frequently they had vaginal or anal sex with each partner, and how often they used a condom. They were also asked if they knew of or suspected that any of their sex partners had other sex partners and if their sex partner(s) HIV status was positive, negative, or unknown. Participants were asked about engaging in transactional sex both in the beginning and later on in the interview. At the beginning, the interviewers stated, “Sometimes people have sex with someone because they’re in a rough situation and need money, food, shelter, drugs, or something else”. They then asked, “In the past 3 months, have you had sex with someone for money, food, clothing, shelter, drugs, or anything else?” In case the participant was uncomfortable answering the initial question, later on the interviewer asked, “A lot of people think about having sex for money, food, or shelter. Have you had similar thoughts?” followed by, “Have you ever had sex in exchange for something? Tell me about that”. For the present study, we focused on discussions related to housing and relevant syndemic factors (see Ref. [15] for interview guides).

### 2.4. Data Analysis

Interviews were transcribed verbatim and then coded and analyzed in NVivo Version 12, (Lumivero, Denver, CO, USA). Initially, transcripts were analyzed utilizing thematic content analysis [39] to systematically identify primary themes. Themes were developed into a preliminary codebook and were further refined across multiple iterations via the team coding of baseline and follow-up interviews. For the present study, the authors re-analyzed data to explore housing insecurity. All interviews (18 participants; total 47 interviews) with information regarding housing were systematically analyzed line-by-line to identify primary themes. Participants’ interviews were analyzed sequentially to explore and identify changes in housing. Participant numbers are used to protect participants’ privacy; interview number, housing category, real ages, and race/ethnicity are reported.

## 3. Results

Participants included 18 BLCW (*M age* = 31.8, *SD* = 7.55, range: 21–47). All participants were mothers of at least one child under 18-years-old (*M* children = 2.92, *SD I* = 1.44, range: 1–6). Ten participants identified as Black/African American, six identified as Latina, and of the two Afro-Latina participants, one identified as Black/African American and the other as Latina (see Table 1).

### 3.1. Structural Factors: Housing, Environment, Discrimination, and Poverty

Four categories of housing experiences emerged: (a) very unstable: experienced homelessness, stayed in more than one shelter, or doubled up with different friends/relatives; (b) unstable: stayed in same shelter or transitional housing; (c) stable substandard: lived in the same residence but with substandard conditions, including corroded appliances, limited dumpsters for the number of residents, noise, pests, limited space, and neighborhood disorder; and (d) stable costly: lived in the same residence throughout the study with no indication of moving, but rent was expensive relative to their income. For each participant, we identified their housing experience prior to the first interview (3 months or more before the first interview) and during the first interview, the second interview, and third interview (see Figure 2).

#### 3.1.1. Very Unstable Housing

Several participants (22%) reported experiencing very unstable housing, particularly before the study (50%), where they reported experiencing homelessness, living in different shelters, and/or doubling up with different friends or relatives. One participant described experiencing homelessness prior to the study:

Cause me and my first son, while I was pregnant, we were homeless and we were sleeping in my car and my car had broke down… After that [family] told me I couldn’t stay with them… [my sister] dropped us off over here [shelter/transitional housing] with all of our bags and stuff.(218, Prior to Interview 1, Very Unstable, 22 yo, Black)

Once participant 218 arrived at the shelter, she was assigned a case manager, who set up housing for participant 218 in a local hotel. Shortly afterward, her case manager assigned her to transitional housing in which she remained for the duration of the study.

#### 3.1.2. Unstable Housing

Some participants (22%) reported that they lived in the same shelter or transitional housing. A participant described her experience in transitional housing, in which she remained throughout the study, and her difficulties finding permanent housing:

Housing should be coming through pretty soon. I applied on different waiting lists and I’m like 50 [on the list]… they made a mistake and they took me off [the list] and put me back on the [bottom] of the list but I got the papers to show that I was already on the list.(212, Interview 1, Unstable, 46 yo, Afro-Latina)

This participant even applied for permanent housing in a rural area about an hour outside of Austin but was again put on a very long wait list. Due to the long wait lists in and around Austin, she was unable to find any permanent housing for her and her young children prior to the end of the study.

#### 3.1.3. Stable Substandard Housing

Many participants (44%) lived in residences with corroded appliances, limited dumpsters for the number of residents leading to trash piles, noise, pests, and a small residence for the size of the family and reported neighborhood disorder, which was particularly concerning regarding their children. Participant 244 described how she felt unsafe at her current residence:

At first it was good… It was like a last [minute] temporary thing because we have a voucher you have to hurry up, they give you a certain amount of months to find somewhere to stay before your voucher [expires]… It was all right, and then until recently, people’s cars have been getting broken into, so now I got to watch outside for my car… But it’s like 10 cars got broken into in two weeks. 10 cars.(244, Interview 2, Stable Substandard, 30 yo, Black)

Participant 244 felt that the housing assistance program placed her in an unsafe neighborhood, and she was eager (but unable) to find safe, affordable housing during the study.

#### 3.1.4. Stable Costly Housing

A couple participants (11%) lived in housing that was expensive and often worked overtime or found side jobs to ensure that rent and utilities were paid. Participant 224 was abruptly evicted from her prior residence. With less than one week to move, she could only find a small apartment that was expensive compared to her previous apartment, where she reported neighborhood disorder and income obtained from her current job:

It’s expensive here and I don’t think it’s worth it, like I don’t. Like if you look at these apartments, if you look at the cabinets and stuff, it’s not worth it. And it’s super small…They’re way overpriced. So, I’ve been really thinking about, once I figure out like a job that I’m secure, I really been thinking about going to [town over 1 h north of Austin] because I’m like for what I’m paying here, I could have a house.(224, Interview 3, Stable Costly, 34 yo, Black)

The cost of housing led Participant 224 to consider moving outside of where she worked. She realized that she could rent a house rather than an apartment for the same amount of money, but that would require her to commute approximately 1.5 h each way.

### 3.2. Social Factors: IPV and Trauma

Participants reported experiencing various forms of IPV, including emotional, physical, sexual, economic, and social isolation. In general, participants who experienced more stable housing across the interviews reported less IPV. All participants reported experiencing at least one form of IPV during their first interviews; however, those reporting very unstable or unstable housing tended to report more IPV during interviews 2 and 3.

#### 3.2.1. Emotional Abuse

Emotional abuse includes non-physical acts by a former or current significant other to control, scare, or isolate an individual [40]. Gaslighting, which causes an individual to question their feelings, instincts, and sanity [41], is also a form of emotional abuse. Generally, participants who experienced more emotional abuse tended to report more adverse housing, particularly those living in very unstable conditions. For example, all participants who lived in very unstable housing reported experiencing emotional abuse across all interviews, yet participants living in stable substandard or stable costly housing reported decreasing emotional abuse across interviews, with only one participant who reported this form of IPV during interview 3 and lived in stable substandard housing (see Table 2). One participant described the emotional abuse she experienced by her partner in their own home:

It’s like he just try to start stuff, or it’s just like when we go out… So his grandma came up there, because we live on the second floor, she came up there with [boyfriend’s] brother and his uncle. So everyone’s sitting around talking, and every time his uncle and his brother… The dudes in his family, the males, [boyfriend says] “you put your head down”. So I say “hello”. “Oh, you didn’t have your head down”. And it’s like his little brother. I’m like this is your… He’s 22, but I’m like this is your little brother, like come on.(244, Interview 2, Stable Substandard, 30 yo, Black)

Participant 244′s partner would not even allow her to look at another man, even if it was his own relative. Prior to the study, participant 244 worked at a daycare. Her partner picked her up one day and saw a father picking up his child. Participant 244′s partner was angry that she was working with a man; when she tried to explain that she did not work with this man, rather he was just picking up his child from daycare, her partner demanded that she quit her job. The emotional abuse she experienced turned into economic abuse as she no longer had an income due to her partner’s demands.

#### 3.2.2. Physical IPV

Physical IPV occurs when a current or former partner hurts or attempts to hurt their partner using violence, such as hitting, kicking, or another form of physical force [42]. Participants who reported very unstable housing reported more physical IPV. All participants reported decreased physical IPV across interviews, but participants living in very unstable housing reported more physical IPV than other housing categories (see Table 2). Participant 216 described her experience of physical violence by her partner:

Participant 216: He slapped, he just, he just slapped me. My face hurts. Because he have a heavy hand.

Interviewer: Doesn’t [domestic violence shelter] have an emergency shelter for situations like that?

Participant 216: Yes, but it’s full.(216, Interview 2, Very Unstable, 24 yo, Black)

She considered going to a domestic violence shelter but was unable to as they did not have enough room. As such, she remained with her boyfriend until her third interview. At that point, he became so physically violent that she called her ex-husband to pick her up and ended up staying with him, their daughter, and his parents, for the duration of the study where she was unwanted.

#### 3.2.3. Sexual IPV

Sexual IPV is when a current or former partner forces or attempts to force their partner to participate in a sexual act, sexual touching, or non-physical sexual event (e.g., sexting, watching pornography) when their partner does not or cannot consent [42]. In general, participants who experienced very unstable and unstable housing tended to report more sexual IPV across interviews (see Table 2). One participant described the sexual coercion she experienced to have a place to stay:

I stayed with ex-boyfriend. I stayed with him. This is why I say I want someone to be respectful, is because I told him, I wanted to come and stay at his place, be friends only and I’ll pay for rent. I’ll find a job and I’ll pay for rent and stuff. If you need help watching your son on days that I’m not working, I can do that. That didn’t happen. I went there. He’s like, “You either be my girlfriend or you get out”. I said, “What the hell?” … I was forced into a relationship that I didn’t want to be in… I had to leave. I got tired of it.(230, Interview 1, Very Unstable, 30 yo, Latina)

Although participant 230 described the situation as her ex-boyfriend demanding she enter a relationship, she also stated that they engaged in sexual activities, indicating that she was in a coercive sexual relationship.

#### 3.2.4. Economic IPV

Economic abuse is when a partner controls their partner’s ability to acquire, use, and/or maintain resources, often leading to financial dependence [43]. Participants reporting stable costly housing reported no economic IPV, those living in unstable and stable substandard housing reported decreasing economic IPV, and those living in very unstable housing conditions reported increasing economic IPV (see Table 2). A participant described a situation in which she was financially dependent on her ex-husband, who also created a situation in which she lost her job:

I was financially dependent on him… He was selling drugs and stuff. And doing a lot of stuff he shouldn’t have been doing. He made me lose my job… Because he would call up to my job. He tried to change my life insurance to his name. That’s why I was like, but I still stayed with him so I don’t know. But at that point I felt like he was going to kill me or something.(212, Interview 1, Unstable, 46 yo, Afro-Latina)

Participant 212 was extremely concerned over the safety of her and her children and eventually left her ex-husband and went to a shelter. Shortly after, she was placed in transitional housing but was unable to secure permanent housing during the study even though she was on wait lists for housing in and outside of Austin.

#### 3.2.5. Social Isolation

An individual may control their partner by attempting to isolate them by controlling their social interactions, limiting their outside involvement, and separating them from their friends and family [44]. Many participants stated that they experienced social isolation, often in concert with other forms of IPV. Although participants who lived in unstable housing reported no social isolation by their partners, those living in stable costly, or stable substandard housing reported decreasing social isolation; participants in very unstable housing reported decreasing and then increasing isolation (see Table 2). Participant 210 described how the social isolation she experienced changed her personality:

I wasn’t myself. I lost my personality. I lost my bubbliness. I lost everything about me so I was basically like a church mouse in the house. I couldn’t go anywhere. He was controlling. Couldn’t go nowhere. He made me drop out of school the first time. He would take my car so I wouldn’t go anywhere. And if I wouldn’t, like if I would have my car keys he would slice my tires so that I couldn’t go anywhere, yeah it was bad, it was bad.(210, Interview 1, Unstable, 31 yo, Black)

Fortunately, she left this partner and moved into a shelter with her children. The new partner she had during the study lived in another shelter nearby. Unfortunately, she remained in the same shelter throughout the study as neither her nor her partner were able to find transitional or permanent housing during the study.

### 3.3. Individual-Level Factors: Substance Use, Mental and Physical Health, and HIV Risk

#### 3.3.1. Substance Use

Participants reported using alcohol, marijuana, and illicit drugs throughout the study. We explored overlaps between housing instability and substance use. Overall, substance use was constant among participants reporting very unstable housing but tended to decrease among all other housing categories.

Alcohol misuse for women is defined as having four or more drinks on a given day, or eight or more drinks in a week [45]. Participants were informed of what a drink serving is and then asked within the past 30 days how many drinks they typically had per week and whether they drank four or more drinks on one occasion. Interestingly, alcohol misuse was low among participants living in very unstable and unstable housing, and alcohol misuse decreased across all housing groups, yet increased slightly among participants living in very unstable housing (see Table 2). One participant described that her drinking increased since her last interview because she was struggling with her ex-husband, and she hid her drinking from her children:

I’ve been lately just drinking left and right. I’ve been drinking whatever I can get my hands on. Budweiser, Coronas… I’m drinking when [the kids are] asleep, because I don’t want them to see me… And I’m still getting up and going to work. Which is horrible, because I’m just… I know I’m not supposed to be drinking but it kind of numbs that feeling that I’m dealing with. But it’s not good. It’s not good because it just makes it… numb. And it just makes me stop thinking of him for a while but then reality really sinks in… And kind of let it out and cry and then by the time I find out I’m like, in bed already… Somehow my subconscious knows that I have to get off the couch, that I can’t leave the bottles on the floor because like, the kids… So I find myself waking up the next morning, and the living room’s clean and like, there’s no sign of me drinking. But I know I drank.(235, Interview 2, Stable Substandard, 29 yo, Latina)

Participant 235 used alcohol to numb her feelings so she would not think about her ex-husband. Eventually, her increased use of alcohol affected her work such that she had difficulty concentrating and made errors that she did not previously make when she was drinking less or no alcohol.

Participants were asked how often they used marijuana in the past 30 days. Those who used marijuana at least 14 of the past 30 days were identified as marijuana users [34]. Across all housing groups, marijuana use decreased across interviews; however, it remained the highest among participants living in very unstable housing (see Table 2). A participant jeopardized her transitional housing by smoking marijuana in her apartment in order to self-medicate:

I started smoking because my head felt like it was gonna to blow up. And then I went to the hospital. They gave me stuff there but it wore off… Next week, on Wednesday I’m going to call and schedule an appointment [with a psychiatrist] if I can get in this week fine. If I can get in next month I get meds. I’m going to stop smoking again. But until I get meds I’m going to keep smoking.(212, Interview 2, Unstable, 46 yo, Afro-Latina)

Participant 212 used marijuana to self-medicate for both pain and depression because she was waiting to see a doctor to be prescribed medication. By the third interview, she reported that she stopped smoking marijuana and was prescribed medication by her mental health provider, yet participant 212 appeared to be overmedicated in that she was extremely quiet rather than her usual bubbly self, and she had difficulty answering basic questions, had difficulty concentrating and processing her thoughts, and complained that her brain felt “foggy”. The interviewer strongly recommended that she contact her mental health provider to inform them of her symptoms since she started the medications.

Illicit substance use is defined as the use of cocaine, crack cocaine, heroin, hallucinogens, inhalants, or methamphetamine or the misuse of prescription medication [46]. Overall, illicit substance use was low across participants in all housing categories, yet participants reporting very unstable housing reported more and consistent illicit substance use than those in other housing categories (see Table 2). When Participant 216 was asked if she ever felt like she was using too many illicit substances, she responded regarding her use of cocaine:

Last week I almost had a heart attack. I was on the couch laying down and I felt like pain in my heart. I’m like, “I need to stop”.

When asked if she went to the hospital, she responded as follows:

No. They put me in a cold shower. [Boyfriend] and his friend put me in a cold shower. Then I blacked out after that I don’t know what happened. Like, [interviewer name], I’m so serious, the last thing I heard from him was “sex stimulation”.

Next, the participant was asked if she believed that her boyfriend had sex with her when she passed out and she responded as follows:

That’s the last thing I heard. I’m like what the fu….then my sister, my sister called my phone and was like “Dude you need to get away from this dude, you really do”. And I’m like, “I don’t know how to”.(216, Interview 3, Very Unstable, 24 yo, Black)

Participant 216 was in a volatile relationship where her boyfriend used and encouraged the use of illicit drugs, which had negative consequences (i.e., participant 216 lost consciousness). However, she was unable to leave this situation as she was dependent on her boyfriend for housing and all local shelters were full.

#### 3.3.2. Risk of Acquiring HIV

Participants reported engaging in sex without a condom that was sometimes coercive, having multiple sex partners, or their main sex partner having multiple sex partners. At interview 1, all participants were at an increased risk of acquiring HIV; however, all housing categories except for participants in very unstable housing, reported a decreased risk of acquiring HIV across interviews.

In their first interview, all participants reported having sex without a condom in the prior 3 months. Overall, participants tended to report fewer sexual encounters without a condom throughout the study, although participants reporting very unstable housing indicated having more condomless sex (see Table 2). One participant explained throughout her three interviews that she would let her sexual partner decide if they used condoms. For example:

Yes [I did want to use a condom]… I didn’t ask him, because I knew he didn’t want to… I wasn’t afraid to ask. I just knew he didn’t want to.(218, Interview 3, Unstable, 22 yo, Black)

Participant 218 indicated her desire to use a condom with one of her sex partners. He previously told her that he did not like using condoms, so rather than requesting to use a condom, she deferred to what she believed this sex partner wanted based on her previously experiences with him.

Each participant was asked about sex partners within the past 3 months from their first interview and since the last interview for interviews 2 and 3. Although the number of sex partners tended to decrease across interviews among most participants, it increased by the third interview among participants in unstable housing (see Table 2). One participant explained the following in her first interview:

I’ve had, oh God. I’ve had about maybe three casuals.(228, Interview 1, Very Unstable, 32 yo, Black)

During her second interview, the same participant expressed that she ended the casual sexual relationships:

I just stopped answering [partner 1′s] phone call. I got a call blocker for text messages and stuff and then with [partner 2] I just told him at school, like, “Hey, work on your marriage… cause it’s not fair to me. I’m always gonna be a secret. I can’t go in public with you”. I didn’t like that. And then with [partner 3], he was a whore so I don’t think it really phased him that I stopped talking to him. I didn’t even give him a reason. I just walked passed him like I didn’t see him. Granted, both these guys [partners 2 and 3] go to my school.(228, Interview 2, Very Unstable, 32 yo, Black)

By her third interview, participant 228 was in a committed, mutually monogamous relationship with her best friend’s brother and looking for permanent housing for them and her son.

Several participants expressed that they knew or were concerned that their main partner was having sex with other partners. Most participants knew of or suspected their sexual partners having multiple sex partners in interview 1, which decreased across interviews in part due to participants leaving romantic or sexual partners or starting new, mutually monogamous relationships. Participants living in unstable housing tended to report knowing or suspecting that their partner(s) were sexually involved with other sex partners more so than participants in other housing categories (see Table 2). One participant was exposed to a sexually transmitted infection (STI) because her sex partner had sex with someone else a few weeks before they were intimate:

[Ex-husband’s] like, “I’m going to try to work things out with you,” and I’m like, okay, let’s see how it works… So that night, I remember we kind of just let it go, I don’t really think much of it… And then [ex-husband says] “I find out that I need to make an appointment because it’s really hurting, it’s being discomfort when I use the restroom”. I got him a doctor’s appointment at the local clinic. He goes in. They test him for STDs. The following day he calls me and tells me, “You know what? I need you to help me go pick up medication,” because he didn’t have a car… I go pick it up, in the back of my head I’m like, “what type of medication did they give him?” I read it. Google what the medication is, and it’s for chlamydia… I go back and the timeline puts him in Mexico. If that’s what it is, it puts him in Mexico, because I had my Pap smear done in early [month] and nothing came back. Nothing came back, so he exposed me to that.(235, Interview 2, Stable Substandard, 29 yo, Latina)

By her third interview, participant 235 reported that her STI test came back negative. She was considering getting back together with her ex-husband and raising their family together in their own home. However, she decided she could no longer trust her ex-husband and chose to remain separated.

Even though this study was observational and did not implement an intervention, we did observe a decrease in many of the variables explored. One participant stated that just describing her lifestyle during interview 1 led her to reflect on her relationships that impacted other variables, such as IPV and the risk of acquiring HIV. She stated the following:

I had this re-evaluation after my [first] interview… I’m like, “You’re very carefree for someone so [sexually] active”.… I was making some really poor decision, like sometimes I wouldn’t use a condom, sometimes I would. I’m just like, “No, if you are STD-free, stay that way”.(228, Interview 2, Very Unstable, 32 yo, Black)

Participant 228 went on to state that she never felt judged during the first interview, but rather merely stating things “out loud” led to her reflecting on certain aspects of her life and ultimately ending sexual relationships with her three partners with whom she inconsistently used condoms. By her second interview, she either ended her sexual relationships or abruptly stopped talking to her former partners, all of whom she reported perpetrated at least one form of IPV. Additionally, by interview 2 she just started a new, mutually monogamous relationship with her best friend’s brother, in which they delayed engaging in sexual intercourse.

## 4. Discussion

This study explored housing insecurity across three months in concert with other syndemic factors, including IPV, substance use, and the risk of acquiring HIV among BLCW in Austin, TX. We prioritized these populations because they experience multiple social vulnerabilities and systemic oppression related to their intersectional characteristics, including race/ethnicity, gender, and motherhood, which are associated with housing insecurity [47,48,49,50,51]. Four categories of housing instability emerged: very unstable, unstable, stable substandard, and stable costly. The longitudinal nature of this study allowed us to explore changes in Structural Factors, such as housing circumstances, along with other syndemic factors, including Social Factors (IPV) and Individual Factors (substance use/risk for acquiring HIV).

Most participants immediately prior to the study reported experiencing very unstable or unstable housing. However, during the 3-month study, all participants who completed all three interviews remained in the same housing categories, which may be in part due to the short length of the study. This may also speak to the difficulty in finding stable, affordable housing as participants in unstable housing lived in the same shelter/transitional housing prior to the study. All participants desired to have different housing circumstances, whether it be larger, more affordable, in a safer neighborhood, or stable so they could leave shelters, transitional housing, or doubling up with relatives. Although this study was conducted based on a small sample size, the voices of participants provide examples of the systemic racism experienced by BLCW in Austin regarding housing insecurity and homelessness. According to conservative and likely underreported data, 40% of individuals experiencing homelessness are women, over a quarter reported IPV, and these data are much higher among BLCW compared to White cisgender women [8,9].

In Austin, TX, there are current initiatives to assist with housing insecurity and homelessness. The Austin City Council adopted a Strategic Housing Blueprint in 2017 [52]. This Blueprint addresses the major factors affecting housing affordability in Austin, and a Displacement Mitigation Strategy was created by the Housing and Planning Department to incorporate into the Blueprint’s Implementation Plan, which presents the City’s focus to meet its goals in the upcoming years [52,53]. The Displacement Mitigation Strategy aims to reduce various types of displacement that often lead to housing insecurity through proactive recommendations that address the needs of the community [53]. When looking at the relationship between HIV and housing instability, Project Transitions, a nonprofit provider of supportive transitional housing for people living with or impacted by HIV and their families, serves the Austin community [54]. Organizations, such as Project Transitions, have been crucial in helping reduce the housing and financial instability populations may face. Thus, there are current policies and initiatives to address housing insecurity and homelessness in Austin. However, there is much that still needs to be done, as all participants described difficult living situations.

Regarding social factors, experiences of various forms of IPV tended to decrease across the three interviews, mainly due to participants ending relationships with romantic or sexual partners who perpetrated IPV. Even with IPV decreasing, participants who experienced very unstable or unstable housing tended to also report more and various forms of IPV. This is confirmed by previous research that also determined that IPV is higher among individuals with more unstable housing [55,56,57]. IPV may be higher among individuals reporting more housing instability because it increases stress, and increases in stress coincide with increased IPV [58,59]. Another possibility is the bidirectional relationship between IPV and insecure housing. Women who experience IPV may depend economically on their partners, which greatly limits housing options after separation. On the other hand, women who experience housing instability due to economic constraints may choose to return to their partner [56,60]. Interestingly, participants tended to only speak about housing in conjunction with IPV when they discussed leaving a partner who perpetrated physical IPV and often went to a domestic violence shelter or that they financially could not afford housing without their partner. Of note, participants were not specifically asked about different types of or categories of IPV, but rather, interviewers asked descriptive questions. Thus, many participants may not have identified that they experienced IPV and as such would not have discussed the link between their IPV experiences and their housing circumstances. Additionally, as mentioned previously, some participants ended romantic or sexual relationships during the study, opting for a new, mutually monogamous relationship. As IPV tends to increase once partners cohabitate as compared to when dating [61], participants involved in new, non-cohabitating relationships may not have experienced IPV until after the study had ended. It is important to note that violence against women is a multifaceted problem. Women who experience poverty are more likely to experience IPV, particularly related to conflict, women’s power, and male identity. More violent relationships tend to have increased conflict regarding finances, jealousy, and women’s “transgressions” related to gender roles. Crises regarding male identity, poverty, and/or inability to control women tend to increase violence against women [62]. Additionally, women and mothers experience violence and structural violence from landlords and housing authorities. For example, landlords have evicted participants with short notice, and housing authorities have continually not met BLCW’s needs and even sabotaged applications by removing them from the list without reason or warning.

Related to individual-level factors, the reported use of alcohol, marijuana, or illicit substances also tended to decrease across interviews but remained higher among BLCW who reported very unstable housing. Other research demonstrated similar findings in that more substance use was associated with more housing instability [63,64,65,66]. Substance use may be used as a way of coping with increased housing instability [67,68]. Additionally, in a previous study, it was observed that among unhoused populations, elevated drug consumption was used as a survival tactic. Individuals may use drugs to curb their hunger when food resources are scarce and to remain vigilant in safeguarding themselves and their possessions [65]. In the present study, participants misused substances for several reasons, including to manage mental and physical health symptoms, and the use of such substances in their residences threatened their housing situations, or they lived with individuals who frequently misused substances and were pressured or coerced to also use substances. Although participants did not always directly link their housing circumstances with substance misuse, their reasons behind using substances were often structural in nature, emphasizing the need for structural- and multi-level interventions, such as expanding insurance coverage and utility, having accessible and affordable mental and physical health providers, and safe, affordable housing opportunities.

The risk of acquiring HIV across interviews due to engaging in vaginal and/or anal sex without a condom, having multiple sexual partners, or participants’ sexual partners having multiple sexual partners slightly decreased, mainly due to participants ending sexual relationships with partners and/or entering into new, mutually monogamous relationships as the study progressed. During interview 1, all participants reported recently engaging in vaginal and/or anal sex without a condom, although this decreased throughout the study, particularly among participants reporting more stable housing. This may be due to the complex interplay between housing insecurity and substance use. Using alcohol or drugs can heighten the probability of individuals participating in condomless sex [69]. According to Dickson-Gomez and colleagues [70], an increased risk of acquiring HIV could be contributed to participants’ unstable or insecure living arrangements with acquaintances or sexual partners, which may entail sharing living expenses and/or substances. This situation can lead to engaging in transactional sex as a means to secure a place to stay [70]. Participants who reported that their sexual partner had other sexual partners generally decreased across the three interviews, mainly due to participants ending relationships with non-monogamous sexual partners. Although most participants did not directly link their risk of acquiring HIV to their housing situations, enquiring about descriptive life situations demonstrated a potential link that may be currently underreported. Future research should explore this further, particularly by considering economic abuse by a sexual partner who has other partners and/or participants engaging in survival sex.

The intersectional identities of BLCW lead to experiences of systemic oppression, including but not limited to machismo, racism, classism, and xenophobia, which increase the likelihood of experiencing housing insecurity and homelessness [71]. These social vulnerabilities must be prioritized and addressed to improve syndemic-related health outcomes among BLCW and their children, including but not limited to increasing affordable housing and expanding housing subsidies to reduce waitlist times, providing rent relief, providing eviction assistance, improving shelters and providing more social services, and educating individuals involved with BLCW and their children experiencing housing insecurity in anti-oppressive practices [72,73,74].

### 4.1. Limitations

This study contributes to the literature by exploring housing insecurity over time in concert with other syndemic factors among BLCW; however, it is not without its limitations. This study was exploratory and qualitative in nature, limiting its generalizability, as this study was conducted among a small sample of BLCW in Austin, TX. Additionally, these data originated from a larger study that was focused on exploring syndemic factors along with the risk of acquiring HIV and the interest in and adoption of PrEP [15]; housing insecurity was not the main focus of the larger study but did emerge as an important syndemic factor experienced by participants. Due to housing insecurity not being the main focus when data were collected, we may have limited saturation to emergent themes pertaining to syndemic theory and housing insecurity among BLCW. Although participants may have underreported stigmatized experiences and behaviors (i.e., IPV, substance use, factors increasing one’s risk of acquiring HIV) due to social desirability, participants tended to report more of these stigmatized experiences and behaviors at interview 1, indicating they may have felt comfortable disclosing these factors. Only one interview was conducted in Spanish; future research should consider recruiting more participants where Spanish is their preferred language.

### 4.2. Future Research

Future research should explore housing insecurity and other syndemic factors in other geographic locations, among other intersectional populations and communities, and among a larger sample size. Insecure housing, IPV, substance use, and the risk of acquiring HIV are often stigmatized; thus, participants may feel more comfortable completing questionnaires regarding their experiences and behaviors that can be followed up with an interview to further explore their responses. This will allow for mediation and/or moderation to be evaluated among the aforementioned variables and for other relevant variables to be included. Other research should include more participants with English as their second language as their experiences may differ from native English speakers. Lastly, researchers should conduct additional formative research to identify methods to implement structural- and multi-level interventions to improve housing among BLCW.

## 5. Conclusions

Housing insecurity is extremely concerning, particularly among BLCW, and was experienced in concert with other syndemic factors, including IPV, substance use, and the risk of acquiring HIV. Overall, participants who reported very unstable or unstable housing tended to also report more and different forms of IPV, the use of more substances, and situations that increased their risk of acquiring HIV. Future research should consider mixed methods and quantitative research to ascertain directionality, as well as consider structural- and multi-level interventions to combat housing insecurity among BLCW through a syndemic theory lens.

## Figures and Tables

**Figure 1 ijerph-20-07177-f001:**
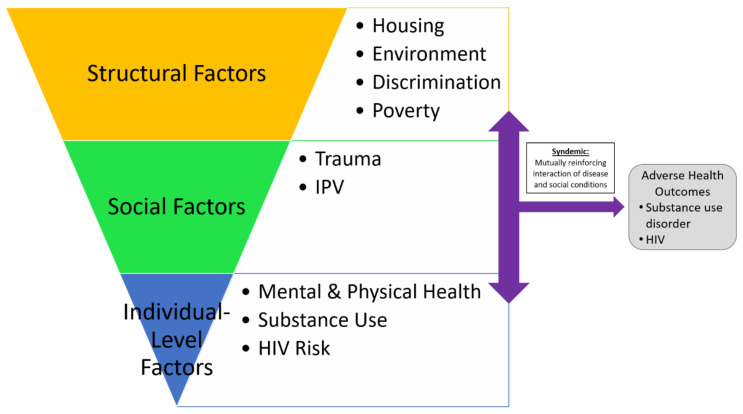
Syndemic theory and factors from a socioecological perspective [16].

**Figure 2 ijerph-20-07177-f002:**
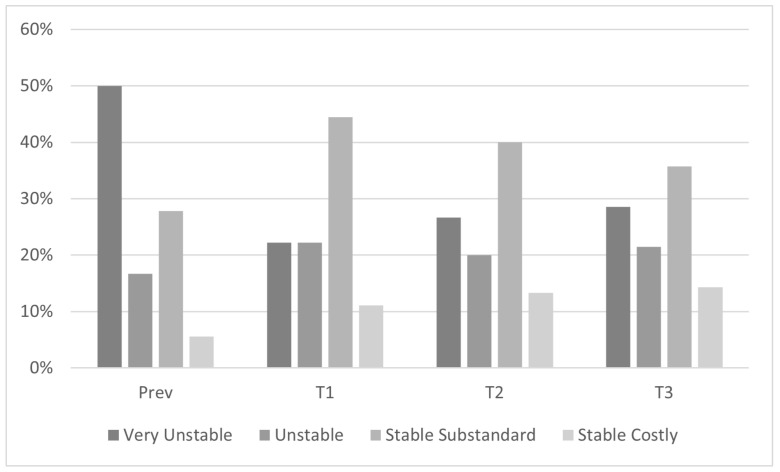
Percentages of housing experiences across three interviews among BLCW in Austin, TX (2018–2019). Four categories of housing experiences experienced by participants prior to and during the study; Prev = prior to the study (*n* = 18); T1 = interview 1 (baseline; *n* = 18); T2 = interview 2 (1-month follow-up; *n* = 15); T3 = interview 3 (3-month follow-up; *n* = 14).

**Table 1 ijerph-20-07177-t001:** Demographic characteristics and syndemic variables of interest among BLCW across three interviews in Austin, TX (2018–2019).

Variable		Total (*n* = 18)	Previous (*n* = 18)	T1 (*n* = 18)	T2 (*n* = 15)	T3 (*n* = 14)
Age (years) (*M* (*SD*))		31.8 (7.55)	-	-	-	-
Number of children (*M* (*SD*))		2.92 (1.44)	-	-	-	-
Race/ethnicity						
	Black/African American	10 (55.56)	-	-	-	-
	Latina/Hispanic	6 (33.33)	-	-	-	-
	Afro-Latina ^a^	2 (11.11)	-	-	-	-
Housing experiences ^b^						
	Very unstable	9 (50.00)	9 (50.00)	4 (22.22)	4 (26.67)	4 (28.57)
	Unstable	6 (33.33)	3 (16.67)	4 (22.22)	3 (20.00)	3 (21.43)
	Stable substandard	9 (50.00)	5 (27.78)	8 (44.44)	6 (40.00)	5 (35.71)
	Stable costly	2 (11.11)	1 (5.56)	2 (11.11)	2 (13.33)	2 (14.29)
IPV ^b^						
	Emotional	16 (88.89)	-	16 (88.89)	8 (53.33)	6 (42.86)
	Physical	11 (61.11)	-	11 (61.11)	1 (6.67)	1 (7.14)
	Sexual	12 (66.67)	-	10 (55.56)	4 (26.67)	3 (21.43)
	Economic	10 (55.56)	-	8 (44.44)	3 (20.00)	2 (14.29)
	Social isolation	11 (61.11)	-	10 (55.56)	2 (13.33)	2 (14.29)
Substance misuse ^b^						
	Alcohol	10 (55.56)	-	9 (50.00)	4 (26.67)	2 (14.29)
	Marijuana	7 (38.89)	-	7 (38.89)	3 (20.00)	1 (7.14)
	Illicit drugs	6 (33.33)	-	6 (33.33)	2 (13.33)	2 (14.29)
Increased risk for contracting HIV ^b^						
	Condomless sex	18 (100.00)	-	18 (100.00)	7 (46.67)	4 (28.57)
	Multiple sex partners	8 (44.44)	-	7 (38.89)	3 (20.00)	3 (21.43)
	Partner has multiple sex partners	14 (77.78)	-	13 (72.22)	5 (33.33)	5 (35.71)

^a^ Of the two Afro-Latina participants, one identified as Black and the other as Latina. ^b^ Not mutually exclusive. Previous = more than 3 months or more prior to the baseline interview. T1 = baseline. T2 = 1-month follow-up. T3 = 3-month follow-up. Results are organized according to the syndemics framework for BLCW at-risk of HIV [16].

**Table 2 ijerph-20-07177-t002:** Percentages of syndemic variables by housing category among BLCW across three interviews in Austin, TX (2018–2019).

Syndemic Variable		Very Unstable			Unstable			Stable Substandard			Stable Costly		
		T1 *n* = 4	T2 *n* = 4	T3 *n* = 4	T1 *n* = 4	T2 *n* = 3	T3 *n* = 3	T1 *n* = 8	T2 *n* = 6	T3 *n* = 5	T1 *n* = 2	T2 *n* = 2	T3 *n* = 2
IPV													
	Emotional	100.0	100.0	100.0	75.0	33.3	33.3	87.5	33.3	20.0	100.0	50.0	0.0
	Physical	75.0	25.0	25.0	50.0	0.0	0.0	62.5	0.0	0.0	50.0	0.0	0.0
	Sexual	75.0	50.0	25.0	50.0	0.0	66.7	50.0	33.3	0.0	50.0	0.0	0.0
	Economic	25.0	25.0	50.0	50.0	0.0	0.0	62.5	33.3	0.0	0.0	0.0	0.0
	Social Isolation	50.0	25.0	50.0	25.0	0.0	0.0	75.0	16.7	0.0	50.0	0.0	0.0
Drugs													
	Alcohol	0.0	0.0	25.0	25.0	33.3	0.0	75.0	50.0	20.0	100.0	0.0	0.0
	Marijuana	50.0	50.0	25.0	50.0	33.3	0.0	37.5	0.0	0.0	0.0	0.0	0.0
	Illicit	25.0	25.0	25.0	25.0	0.0	0.0	37.5	0.0	0.0	0.0	0.0	0.0
Risk of acquiring HIV													
	Condomless sex	100.0	50.0	75.0	100.0	33.3	33.3	100.0	50.0	0.0	100.0	50.0	0.0
	Multiple sex partners	50.0	25.0	0.0	50.0	33.3	66.7	37.5	16.7	20.0	0.0	0.0	0.0
	Partner with multiple sex partners	100.0	25.0	25.0	100.0	66.7	66.7	50.0	16.7	20.0	50.0	50.0	50.0

T1 = baseline. T2 = 1-month follow-up. T3 = 3-month follow-up. IPV = intimate partner violence. Results are organized according to the syndemics framework for BLCW at-risk of HIV [16].

## Data Availability

The data presented in this study are openly available in the Texas Data Repository at https://dataverse.tdl.org/dataset.xhtml?persistentId=doi:10.18738/T8/X61LJB (accessed on 31 October 2021).

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
