# Peer review of "Housing Insecurity and Other Syndemic Factors Experienced by Black and Latina Cisgender Women in Austin, Texas: A Qualitative Study"

_ijerph, 2023, doi:10.3390/ijerph20247177_

Round 1
Reviewer 1 Report
Comments and Suggestions for Authors
The paper deals with housing insecurity and other syndemic factors (poverty, environment, discrimination) among Black and Latina cisgender women (BLCW) in Austin, Texas. Austin, being one among the fastest growing cities in the US, is very specific environment for BLCW. The sample used in the research is rather small: 18 participants. However, such a sample is justified, since in-depth, semi-structured interviews were used. The paper identifies four categories of hausing insecurity. In terms of methodology, the paper is reliable and raises no objections. The findings of research are, however, limited. The Authors are right when stating that “future research should consider recruiting more participants where Spanish is their preferred language” (p. 17).
Author Response
Thank you for your review and understanding of our limitations and identification for much-needed future research.
Reviewer 2 Report
Comments and Suggestions for Authors
The authors tackle a very relevant subject with a closeness and sensitivity appropriate to the situation. I congratulate them on their study. I would like to point out the following considerations to contribute to the writing of the paper:
INTRODUCTION:
1. Include the concept of housing insecurity in the introduction.
2. Include a paragraph with the factors associated with the profile of homeless people already identified in the literature, citing the references.
OBJECTIVE AND JUSTIFICATION OF THE STUDY:
1. Develop the justification for the study by retrieving some data on the high prevalence of homeless people and the main health consequences of living on the streets. This will make the justification more consolidated.
2. Make the objective more succinct, make the main object of study clear. The current form gives the impression that the study has many objects of study (housing insecurity, intimate partner violence, breast cancer, substance use, risk of contracting HIV). Even though we understand that there are syndemic factors, they are secondary outcomes to the central object of study.
METHODS:
1. The study claims to have a qualitative approach, but in the results the quantitative approach stood out, with many figures containing descriptive quantitative data.
2. Include in the "methods" section how it was defined that women are at risk of contracting HIV.
3. Identifying the participant's preliminary eligibility by telephone is already a factor that excludes women who don't have access to a telephone. Identify this in the exclusion factors.
4. Why is it necessary for the participant to have at least 1 child? This is not related to the aim of the study.
5. Briefly explain why you chose the particular BLCW data collection and recruitment sites to approach participants. Are homeless people concentrated in these locations for any particular reason?
6. There's no need to put the author's name in the middle of the text, just identify it indirectly. For example: "... interview room in the office of one of the study's authors"... (example) or "... interview room in the medical office of one of the authors of the study"
7. There's no need to put the author's name in the middle of the text, identify her indirectly, or state her main training/graduate degree. For example: "Most of the interviews were conducted by a psychologist (example), trained in qualitative methods...."
8. Indicate how the interviews were recorded. Did they use a microphone or camcorder?
9. To make it clear how it was analyzed, I suggest identifying exactly the points that were asked and how they were measured, so that the survey really has the principle of replication.
10. Did the study have a guiding question?
11. What is the main theoretical framework used to analyze the housing insecurity of the participants?
RESULTS:
1. Complete the title of the table with key information about the study, so as not to generalize the data in the table. For example: "Characteristics of homeless BLCW women according to demographics, life experience and health problems. Austin - Texas, 2018 and 2019. (n=18).
2. Tell us how the housing experience was measured. How the classification of Very unstable; unstable; stable below standard basically occurs.... Describe this in the methodology, identifying the theoretical framework (Did the authors themselves create this housing classification?)
3. Check the limit of figures and tables informed by the magazine. I suggest you check if you can combine the data from figures 3 to 13 and turn it into one or two tables.
4. If possible, do not write the caption in the same field as the figure title.
5. Figures 3 to 13: complete the title of the figure with key information about the public studied. So as not to induce generalization of the data in the figure.
6. What the role of the case manager was, since he intervened in one of the cases (participant 218). Describe this in the methodology.
7. Figures 3 to 13: Describe in the body of the text the most relevant data from all the figures presented. It is not intuitive to look at the figure and abstract the main data that the authors want to highlight.
8. The excerpts from the participants' speeches are large and can be distracting from the point the authors want to highlight. I suggest identifying and keeping the central speech, making it more succinct. It will be clearer what the authors want to highlight.
DISCUSSION:
1. I suggest that the discussion be held with the presentation of the results, it could increase the fluidity of the text.
2. I suggest including in the discussion a section on the public policies surrounding the problems observed.
CONCLUSION:
1. The conclusion appears to be more quantitative than qualitative. Relating to which group had a higher frequency of syndemic factors.
REFERENCES:
1. I suggest reviewing the rules for writing references. Some references are missing information, some only have the title of the work and the access link. There is no author, year of publication, publisher....
2. Check the possibility of updating some references.

Author Response
INTRODUCTION:
- Include the concept of housing insecurity in the introduction.
Response: This was added to the introduction.
- Include a paragraph with the factors associated with the profile of homeless people already identified in the literature, citing the references.
Response: Based on another reviewer’s comments, we added information under 1.2. that specifies factors experienced by people experiencing homelessness or housing insecurity.
OBJECTIVE AND JUSTIFICATION OF THE STUDY:
- Develop the justification for the study by retrieving some data on the high prevalence of homeless people and the main health consequences of living on the streets. This will make the justification more consolidated.
Response: We added the justification including stats on homelessness among women and health consequences of being homeless
- Make the objective more succinct, make the main object of study clear. The current form gives the impression that the study has many objects of study (housing insecurity, intimate partner violence, breast cancer, substance use, risk of contracting HIV). Even though we understand that there are syndemic factors, they are secondary outcomes to the central object of study.
Response: We clarified that the main objective so that the main focus is on housing.
METHODS:
- The study claims to have a qualitative approach, but in the results the quantitative approach stood out, with many figures containing descriptive quantitative data.
Response: We agree and removed figures 3-13 to focus on qualitative findings. We indicated the general trends that we previously had in figures 3-13 for each section.
- Include in the "methods" section how it was defined that women are at risk of contracting HIV.
Response: We originally stated that this information is provided elsewhere as this manuscript was not specific to HIV. Given the reviewer’s request, we added additionally eligibility criteria.
- Identifying the participant's preliminary eligibility by telephone is already a factor that excludes women who don't have access to a telephone. Identify this in the exclusion factors.
Response: In 2.1, we added the following sentence: “Exclusion criteria included 1) being less than 18 years old, 2) identifying as any race or ethnicity aside from Black/African American or Latina/Hispanic, 3) identifying as any other gender than cisgender woman, 4) having no children or only children older than 18 years old, 5) using condoms during every vaginal or anal sexual encounter with a cisgender man in the past 30 days, 6) living with HIV, 7) only spoke fluently in any language other than English or Spanish, and 8) having no access to a phone to determine preliminary eligibility.” While we made this change based on the reviewer’s comments, since this manuscript is an analysis of a larger study, we’re concerned that the continued focus on HIV for this current manuscript is distracting from the main point of this manuscript.
- Why is it necessary for the participant to have at least 1 child? This is not related to the aim of the study.
Response: Correct, it is not related to the aim of the study. As mentioned under 2.1. Participants, the eligibility criteria of having at least 1 child under 18 was part of the larger study.
- Briefly explain why you chose the particular BLCW data collection and recruitment sites to approach participants. Are homeless people concentrated in these locations for any particular reason?
Response: As mentioned previously, the larger study was on HIV prevention and PrEP. During interviews and while analyzing data, it was clear that housing instability was a major issue among participants. We did not specifically recruit participants based on housing circumstances, it just so happened that participants at high risk for contracting HIV also experienced housing instability. In 2.2, we added the sentence, “We used these sites to recruit BLCW from a variety of places to ensure a diverse background of participants.”
- There's no need to put the author's name in the middle of the text, just identify it indirectly. For example: "... interview room in the office of one of the study's authors"... (example) or "... interview room in the medical office of one of the authors of the study"
Response: We respectfully request the editor to comment on this suggestion. In previous publications where we reported similar information, editors preferred that we stated exactly who was involved by using the author’s initials. If the current editor agrees with the reviewer’s comments, we are more than happy to remove where we’ll insert initials and keep the sentences more vague.
- There's no need to put the author's name in the middle of the text, identify her indirectly, or state her main training/graduate degree. For example: "Most of the interviews were conducted by a psychologist (example), trained in qualitative methods...."
Response: As mentioned previously, inserting the author’s initials to identify them is common practice, particularly for qualitative research. Additionally, we used the “Consolidated criteria for reporting qualitative research” (COREQ) checklist to write this manuscript. Part of this checklist includes identifying interviewers’ training to ensure they were qualified to conduct interviews. Again, I respectfully ask the editor to comment on this–we are happy to remove this information if requested.
- Indicate how the interviews were recorded. Did they use a microphone or camcorder?
Response: In the second paragraph in 2.2 we stated that interviews were audio-recorded, indicating that no camcorder was used. We used 2 audio recorders in the unlikely case one failed.
- To make it clear how it was analyzed, I suggest identifying exactly the points that were asked and how they were measured, so that the survey really has the principle of replication.
Response: We are unclear on how to respond to this comment. As mentioned in the manuscript (i.e., 2.2. paragraph 2), we collected data using in-depth, semi-structured interviews. As such, there is no survey to replicate. The interview guides were published in our previous manuscript (15–blind for review), so readers can refer to those guides to replicate the study. Additionally, the redacted interview transcripts are available in the provided link to the data repository.
- Did the study have a guiding question?
Response: Yes, as stated in 1.2. Study Purpose, this particular manuscript and analysis explored housing experiences in-depth and across time among BLCW. As this was a qualitative study, we explored the data on housing rather than going in with a specific hypothesis, which would be inappropriate for a qualitative study.
- What is the main theoretical framework used to analyze the housing insecurity of the participants?
Response: As mentioned in 1.1. Syndemic Model, we used syndemic theory to guide analyses. In Figure 1, we present a model that we used to analyze our data. The results are structured according to syndemic theory and Figure 1 where 3.1 discusses structural factors, 3.2 discusses social factors, and 3.3 discusses individual-level factors.
RESULTS:
- Complete the title of the table with key information about the study, so as not to generalize the data in the table. For example: "Characteristics of homeless BLCW women according to demographics, life experience and health problems. Austin - Texas, 2018 and 2019. (n=18).
Response: We thank the reviewer for this suggestion. As not all participants experienced homelessness or health problems, we revised the title of Table 1 to, “Demographic characteristics and syndemic variables of interest among BLCW across 3 interviews in Austin, TX (2018 - 2019).”
- Tell us how the housing experience was measured. How the classification of Very unstable; unstable; stable below standard basically occurs.... Describe this in the methodology, identifying the theoretical framework (Did the authors themselves create this housing classification?)
Response: As this was a qualitative study, we did not have specific housing measures. As mentioned in 3.1, after qualitative analyses, four categories of housing experiences emerged. How we categorized housing is detailed in 3.1. This cannot be added to the methodology as this was not determined prior to conducting the study–it emerged from the study transcripts.
- Check the limit of figures and tables informed by the magazine. I suggest you check if you can combine the data from figures 3 to 13 and turn it into one or two tables.
Response: Thank you for your suggestion. We did not see a figure or table limit, but we decided because it is a qualitative study that we should remove the figures. While we considered creating a table, we thought this may also distract from the qualitative findings and instead added more information to each paragraph to inform readers of the overall findings.
- If possible, do not write the caption in the same field as the figure title.
Response: We did this according to the author instructions provided by the journal. We are happy to make changes if these were done incorrectly.
- Figures 3 to 13: complete the title of the figure with key information about the public studied. So as not to induce generalization of the data in the figure.
Response: Thank you for your suggestion; we decided to remove figures 3 to 13.
- What the role of the case manager was, since he intervened in one of the cases (participant 218). Describe this in the methodology.
Response: We are merely reporting what the participant informed us of after the provided quote, so we did not provide an even longer quote. The case manager did not intervene during the study. They assisted participant 218 after she was left homeless by her family. We did expand on the situation after the quote to clarify the role her case manager played in getting her into transitional housing: Once participant 218 arrived at the shelter, she was assigned a case manager, who set up housing for participant 218 in a local hotel. Shortly afterward, her case manager assigned her to transitional housing in which she remained for the duration of the study.
- Figures 3 to 13: Describe in the body of the text the most relevant data from all the figures presented. It is not intuitive to look at the figure and abstract the main data that the authors want to highlight.
Response: We agree; we removed figures 3 to 13 and added text regarding the most relevant data/findings.
- The excerpts from the participants' speeches are large and can be distracting from the point the authors want to highlight. I suggest identifying and keeping the central speech, making it more succinct. It will be clearer what the authors want to highlight.
Response: Thank you for this suggestion, we cut down the text from most of the quotes.
DISCUSSION:
- I suggest that the discussion be held with the presentation of the results, it could increase the fluidity of the text.
Response: Thank you for this suggestion; we revised the Discussion to include more results and synthesis.
- I suggest including in the discussion a section on the public policies surrounding the problems observed.
Response: We added a paragraph and citations at the end of the Discussion.
CONCLUSION:
- The conclusion appears to be more quantitative than qualitative. Relating to which group had a higher frequency of syndemic factors.
Response: We revised the conclusion so it demonstrates the qualitative nature of this study and discusses future research.
REFERENCES:
- I suggest reviewing the rules for writing references. Some references are missing information, some only have the title of the work and the access link. There is no author, year of publication, publisher....
Response: We apologize for not noticing this and thank Reviewer 2 for bringing this to our attention. As suggested in the manuscript template, we used the journal’s pre-determined reference formatting in Zotero. Rather than using Zotero for the revision, we updated the reference list manually.
- Check the possibility of updating some references.
Response: Based on Reviewer 2’s and other Reviewers’ comments, we added several more citations that are more recent.
Reviewer 3 Report
Comments and Suggestions for Authors
This paper explores the experience of housing insecurity and other syndemic factors among Black and Latina cisgender women (BLCW) in Austin, Texas. The Introduction offers a concise background and a well-reasoned justification for the chosen focus and the contribution this study aims to make. As the authors acknowledge, this is a much-needed study given the dearth of focus on housing among BLCW through a syndemic lens.
The study is both methodologically and ethically sound, and follows careful procedures which are tailored to the participant group. I found the practice of allowing participants choice over interview location and the decision to compensate candidates for their time even if they were ineligible for the full study inspiring, for instance. I have only a small number of suggested revisions:
1. I was interested to hear more about some of the details of the participants' experiences, if it is possible to elaborate on these. For example, line185-6 (p.6), "This participant even applied for permanent housing in rural areas but was unable 185 to find anything for her and her young children". Could the authors say anything further/in particular about the participant's difficulty in finding housing for her and her children?
2. I would find it helpful to have more (brief) context provided in the Introduction section about the way the housing system works in Austin, TX. This may be particularly useful for international readers. What kind of assistance is available in terms of housing support?
3. It would be helpful to explain how the label of 'stable/unstable' was defined in the categories assigned to housing experiences. When was a housing experience determined as stable? I would question, for example, whether a housing experience could be classed as stable if the housing is unaffordable. Some further elaboration on these definitions/categorisations would be worthwhile.
4. I wondered whether 'romantic partner' is the most appropriate term when experiences of emotional abuse are involved.
Author Response
The study is both methodologically and ethically sound, and follows careful procedures which are tailored to the participant group. I found the practice of allowing participants choice over interview location and the decision to compensate candidates for their time even if they were ineligible for the full study inspiring, for instance. I have only a small number of suggested revisions:
- I was interested to hear more about some of the details of the participants' experiences, if it is possible to elaborate on these. For example, line185-6 (p.6), "This participant even applied for permanent housing in rural areas but was unable 185 to find anything for her and her young children". Could the authors say anything further/in particular about the participant's difficulty in finding housing for her and her children?
Response: We agree and provided more information after most quotes. For example, for the participant mentioned in Reviewer 3’s comment, we revised to: This participant even applied for permanent housing in a rural area about an hour outside of Austin, but was again put on a very long waiting list. Due to the long waiting lists in and around Austin, she was unable to find any permanent housing for her and her young children prior to the end of the study.
- I would find it helpful to have more (brief) context provided in the Introduction section about the way the housing system works in Austin, TX. This may be particularly useful for international readers. What kind of assistance is available in terms of housing support?
Response: We added 3 sentences to the introduction on what is available in Austin.
- It would be helpful to explain how the label of 'stable/unstable' was defined in the categories assigned to housing experiences. When was a housing experience determined as stable? I would question, for example, whether a housing experience could be classed as stable if the housing is unaffordable. Some further elaboration on these definitions/categorisations would be worthwhile.
Response: Thank you for this comment; we agree that it’s not clearly defined. We revised the category to be: d) stable costly: lived in the same residence throughout the study with no indication of moving but rent was expensive relative to their income. We made those changes throughout the manuscript and clarified where necessary. We also made other clarifications in the results.
- I wondered whether 'romantic partner' is the most appropriate term when experiences of emotional abuse are involved.
Response: Most definitions we cited use the term “romantic partner” to identify individuals involved with a partner and may or may not be sexual in nature. However, we agree that this term is problematic and changed it under “Emotional Abuse” to now read “significant other.”
Reviewer 4 Report
Comments and Suggestions for Authors
The paper addresses the interconnectedness of housing insecurity and syndemic factors such as intimate partner violence (IPV), substance use, HIV risk among Black and Latina cisgender women in Austin, TX. Through a longitudinal study, the authors show how the lack of safe and stable housing is closely tied to higher instances of IPV, increased substance use, and an elevated risk of HIV infection among these women. The study employs a qualitative approach and is based on a series of interviews conducted over three months to explore changes in the women’s housing situations and their connections to changes in other syndemic factors. This research contributes to the existing literature by providing insights into housing instability and its connection to other factors shaping the life experiences and health outcomes of Black and Latina women in Austin, TX.
The manuscript is written in a clear and well-structured manner and presents important qualitative data on housing insecurity among Black and Latina women in Austin, TX. This city has been experiencing a substantial growth and a rising cost of living. The study is based on the syndemic theory, which has been proven effective in analyzing how health and social problems interact and mutually reinforce one another. Such understanding is needed to inform effective interventions that consider the underlying social and economic factors when addressing health issues experienced by marginalized and underserved populations. In this context, the paper contributes to the literature be reinforcing the interconnectedness between health and social conditions, with a specific focus on Black and Latina cisgender women.
The following comment is a suggestion to the authors rather than a required change. I suggest that the paper could benefit from this change, but I am not familiar with the data set and, therefore, cannot be sure if it is feasible. Rather than primarily establishing connections between a lack of stable housing and the increased risk of IPV and other variables (which have been well documented in prior research, as the authors acknowledge with appropriate references to the literature), the interview data could be used to provide deeper insights into how these women personally perceive the links between their housing instability, drug use, risky sexual behaviors, and violence they encounter in their lives. Do they draw parallels between these issues? How do they negotiate these connections in their daily lives? How do they attempt to resolve them? The authors mention some strategies (such as ending relationships with partners, etc.), but they do not expand on these strategies in terms of their connections to housing or the women’s reflections on the issues. Since the primary strength of qualitative research is in providing insights into individual experiences rather than identifying patterns, implementing these changes could make this study more engaging for readers and contribute to its originality. For instance, in the discussion section, the authors use their data to confirm the connections between unstable housing and other factors explored in the study (IPV, drug use, risk of HIV). However, the explanations of why and how these issues are connected are largely drawn from previously published research rather than from the interviews conducted with the women in Austin as part of this study. Integrating this suggestion would likely increase the manuscript’s value, its novelty, and its impact on this field of study.
I have several specific comments:
1. Lines 45-47: “Over one-third (37%) of Black/African Americans and 32% of Latino/as experienced homelessness [7].”
These figures appear to be very high. Can the authors please verify their accuracy? The hyperlink provided in reference 7 directs to a website with a lot of data, but I was unable to locate these precise numbers on the website. Is it possible that the intended statement is that 37% of people experiencing homelessness in Austin were Black/African Americans (and not that 37% of B/AAs in Austin were homeless, as it currently reads)?
2. Lines 217-218: “In general, participants who experienced more stable housing, or more stable housing across interviews, reported less IPV”
Not sure why “more stable housing” is repeated twice in the sentence, it is not very clear.
3. It is unclear from the manuscript whether the women interviewed for this study were involved in an intervention or if it was an observational study (there is a reference in the manuscript, but it is currently blinded for review). Even if briefly, it would be beneficial to provide some details in this regard. If there was no intervention, what would be a possible explanation for why most variables improved over time? Additionally, providing information on the criteria used to determine the number of interviews and the specific choice of 1- and 3-month time intervals would be helpful.
4. Interview guides: what were the major differences between interview guides 1, 2, and 3 (mentioned in line 118)?
5. Is it correct that unstable housing/experience of homelessness was not an eligibility criterion for participants? Were individuals who reported stable housing excluded from the analysis? Total number of participants in the baseline interview was n=18 (T1), but it appears that only 16 responses are reported in the “Housing experience” section in Table 1.
Author Response
The following comment is a suggestion to the authors rather than a required change. I suggest that the paper could benefit from this change, but I am not familiar with the data set and, therefore, cannot be sure if it is feasible. Rather than primarily establishing connections between a lack of stable housing and the increased risk of IPV and other variables (which have been well documented in prior research, as the authors acknowledge with appropriate references to the literature), the interview data could be used to provide deeper insights into how these women personally perceive the links between their housing instability, drug use, risky sexual behaviors, and violence they encounter in their lives. Do they draw parallels between these issues? How do they negotiate these connections in their daily lives? How do they attempt to resolve them? The authors mention some strategies (such as ending relationships with partners, etc.), but they do not expand on these strategies in terms of their connections to housing or the women’s reflections on the issues. Since the primary strength of qualitative research is in providing insights into individual experiences rather than identifying patterns, implementing these changes could make this study more engaging for readers and contribute to its originality. For instance, in the discussion section, the authors use their data to confirm the connections between unstable housing and other factors explored in the study (IPV, drug use, risk of HIV). However, the explanations of why and how these issues are connected are largely drawn from previously published research rather than from the interviews conducted with the women in Austin as part of this study. Integrating this suggestion would likely increase the manuscript’s value, its novelty, and its impact on this field of study.
Response: Thank you for your comment/suggestion. While we expanded on contexts after quotes throughout the Results, we added quite a bit throughout the Discussion to identify where participants made these direct connections, as well as where we made connections based on circumstances that occurred at the same time.
I have several specific comments:
- Lines 45-47: “Over one-third (37%) of Black/African Americans and 32% of Latino/as experienced homelessness [7].”
These figures appear to be very high. Can the authors please verify their accuracy? The hyperlink provided in reference 7 directs to a website with a lot of data, but I was unable to locate these precise numbers on the website. Is it possible that the intended statement is that 37% of people experiencing homelessness in Austin were Black/African Americans (and not that 37% of B/AAs in Austin were homeless, as it currently reads)?
Response: Clarified and rewrote the sentence so it reads more clearly: In 2022, 7.7% of Black/African American and 33.1% of Latino/a communities comprised the population of Austin, TX [8]. Of those who experienced homelessness in Austin, over one-third (37%) were Black/African Americans and 32% were Latino/as [9]. Of individuals experiencing homelessness, 40% were women, 28.3% experienced IPV, and 19.9% were children [10]. Nationally, among individuals experiencing homelessness, 39.4% were Black/African American, 22.5% were Latino/a, 38.5% were women, and 18.3% were children [11]. Although there are low populations of Black/African Americans in Austin, these communities experience high rates of homelessness; and Latino/a communities and women and children experience higher rates of homelessness in Austin than the U.S.
- Lines 217-218: “In general, participants who experienced more stable housing, or more stable housing across interviews, reported less IPV”
Not sure why “more stable housing” is repeated twice in the sentence, it is not very clear.
Response: We deleted the repetition of “more stable housing”.
- It is unclear from the manuscript whether the women interviewed for this study were involved in an intervention or if it was an observational study (there is a reference in the manuscript, but it is currently blinded for review). Even if briefly, it would be beneficial to provide some details in this regard. If there was no intervention, what would be a possible explanation for why most variables improved over time? Additionally, providing information on the criteria used to determine the number of interviews and the specific choice of 1- and 3-month time intervals would be helpful.
Response: Thank you for these comments. We added several sentences to the beginning of the Methods section to clarify the comments that Reviewer 4 mentioned: This study is an analysis of a larger qualitative study exploring barriers to pre-exposure prophylaxis (PrEP) among BLCW who were at high risk for acquiring HIV [15]. Syndemic theory guided the development of study materials. During analysis of the larger study, we determined that housing insecurity was frequently reported by participants and thus is the focus of this manuscript. We conducted longitudinal, in-depth, semi-structured interviews at three separate assessment points (baseline, 1-, and 3-months) among 18 BLCW (M age = 31.8; SD = 7.55; range 21 – 47 years) at-risk for acquiring HIV from May 2018 to November 2019. The main, larger study was a formative, observational study to ascertain syndemic factors that impacted risk for acquiring HIV and PrEP interest and adoption to develop future interventions. The baseline interview guide explored syndemic factors both prior to and currently experienced by participants; interview guide 2 explored changes in syndemic factors from the baseline interview; and interview guide 3 explored changes in syndemic factors from the 1-month interview as well as suggested future interventions. At the end of the first interview, participants were provided information about PrEP and PrEP providers. We interviewed participants again one month later to determine if they sought out a PrEP provider, and three months later to determine if they adopted PrEP and if not, to identify syndemic barriers (see [15] for results regarding PrEP, barriers, and suggested interventions).
Regarding the changes in variables without an intervention, we added a quote and description as to why one participant distanced herself from sexual partners in interview 1, thus decreased her experiences of IPV and risk for acquiring HIV:
Even though this study was observational and did not implement an intervention, we did observe a decrease in many of the variables explored. One participant stated that by just describing her lifestyle during interview 1 led her to reflect on her relationships that impacted other variables such as IPV and risk for acquiring HIV. She stated:
I had this re-evaluation after my [first] interview… I’m like, “You’re very carefree for someone so [sexually] active.”… I was making some really poor decision, like sometimes I wouldn’t use a condom, sometimes I would. I’m just like, “No, if you are STD-free, stay that way.” (228, Interview 2, Very Unstable, 32 yo, Black)
Participant 228 went on to state that she never felt judged during the first interview, but rather merely stating things “out loud” led to her reflecting on certain aspects of her life and ultimately ending sexual relationships with her three partners with whom she inconsistently used condoms. By her second interview, she either ended her sexual relationships or abruptly stopped talking to her former partners, all of home she reported perpetrated at least one form of IPV, and just started a new, mutually monogamous relationships with her best friend’s brother, in which they delayed engaging in sexual intercourse.
Additionally, we added potential explanations throughout the discussion for each of the factors explored.
- Interview guides: what were the major differences between interview guides 1, 2, and 3 (mentioned in line 118)?
Response: As mentioned in the previous response, we added the differences between the interview guides: The baseline interview guide explored syndemic factors both prior to and currently experienced by participants; interview guide 2 explored changes in syndemic factors from the baseline interview; and interview guide 3 explored changes in syndemic factors from the 1-month interview as well as suggested future interventions.
- Is it correct that unstable housing/experience of homelessness was not an eligibility criterion for participants? Were individuals who reported stable housing excluded from the analysis? Total number of participants in the baseline interview was n=18 (T1), but it appears that only 16 responses are reported in the “Housing experience” section in Table 1.
Response: Correct, housing insecurity was not an eligibility criteria as the larger study was focused on syndemic factors experienced by BLCW who were at high risk for acquiring HIV. We clarified in the Methods section that the eligibility criteria reported was for the larger study. Since housing insecurity emerged during data analysis as a syndemic factor, the focus of this manuscript is on housing. No participants were excluded from analysis. And thank you for noticing that Table 1 was incorrect. We updated sample numbers and percentages throughout.
Round 2
Reviewer 2 Report
Comments and Suggestions for Authors
I congratulate the authors on their careful adjustments. I have listed a few considerations to help clarify and deepen the study:
METHODS: There is no need to put the author's name in the middle of the text, identify it indirectly.
METHODS: Indicate how the interviews were recorded. Did they use a microphone or camcorder? [They probably recorded the call.]
METHODS: Identify the main question of the study. What was the question about housing and syndemic factors? The authors did not specify the guiding question of the study.
METHODS: How was the occurrence of intimate partner violence measured? [Describe this in the methodology. This consolidates the forms of measurement adopted in the study. It makes it clear how the data was obtained].
METHODS: I suggest that you describe the structure of the points asked in the questionnaire used by the interviewers, without asking the reader to read reference "15". This will avoid doubts about what data was actually asked in order to obtain the results of this study. If you read the other study (reference 15), you may be asked about data that was not discussed in this study.
RESULTS: I suggest the authors identify the data. When comparing that one group suffered more sexual violence than another, it's important to provide the numerical data. Briefly describe how many percent more one group was assaulted than the other.
DISCUSSION: In the discussion, explore the social vulnerabilities surrounding the characteristics of the study population (WOMAN, BLACK, LATINO, CISGENERO), specifying the systems of oppression (for example: machismo, racism, xenophobia) that condition people with these characteristics to be in an unstable housing situation.
DISCUSSION: Make it clear in the discussion that violence is a multifaceted problem. Some results linking violence to unstable housing seem to restrict the factors that drive violence too much.
Reviewer 4 Report
Comments and Suggestions for Authors
The authors responded to all the comments and made appropriate changes. I don't have any further comments.
Author Response
Thank you very much for taking the time to review this revised manuscript.